



# A Light-Weight Holographic Imager for Cloud Microphysical Studies from an Untethered Balloon

Thomas Edward Chambers[1], Iain Murray Reid[1,2], and Murray Hamilton[1]

[1]School of Physics, Chemistry and Earth Sciences, and Institute for Photonics and Advanced Sensing, University of Adelaide, Adelaide, SA 5005, Australia
[2]ATRAD Pty. Ltd., 154 Ashley St., Underdale 5032, Australia

**Correspondance:** Thomas Edward Chambers (thomas.chambers@adelaide.edu.au) and Murray Hamilton (murray.hamilton@adelaide.edu.au)

**Abstract.** We describe the construction and testing of an in situ cloud particle imager based on digital holography. The instrument was designed to be low cost and light weight for vertical profiling of clouds with an untethered weather balloon. This capability is intended to address the lack of in situ cloud microphysical observations that are required for improving the understanding of cloud processes, calibration of climate and weather models, and validation of remote sensing observation
methods.

From a balloon sounding through multiple bands of cloud, we show that we can retrieve shape information and size distributions of the cloud particles as a function of altitude. Microphysical retrievals from an imaging satellite are compared to these in situ observations and significant differences are identified, consistent with those identified in prior evaluation campaigns.

## 1  Introduction

Clouds play a key role in the hydrological cycle, are a major factor in extreme weather events, have implications for aviation safety (Gultepe et al., 2019) and ground-based astronomy (Hahn et al., 2014), and a lack of understanding of clouds and precipitation has been identified as the leading source of uncertainty in climate and weather modelling (Forster et al., 2021). More detailed observations of the particle sizes, shapes, and thermodynamic phases will help resolve these uncertainties (Morrison
et al., 2020).

In this paper we present a light-weight cloud particle imaging instrument, using digital holography (Schnars and Jüptner, 1994), which is carried on a sounding balloon. In addition we show images of cloud particles from a sounding performed in heavy overcast conditions with mostly liquid-water clouds. Particle concentrations and histograms of cloud particle sizes are measured.



A range of in situ observational techniques, each having advantages and limitations (Baumgardner et al., 2017), is currently employed to determine cloud particle sizes and shapes. Impaction instruments (MacCready and Todd, 1964) allow very high resolution studies of ice crystal surface properties, but are less suited to large ice crystal measurements due to shattering. Optical scattering probes provide useful measurements, and their biases and uncertainties have been well characterised (Baumgardner et al., 2017). Their application to aspherical particle measurements is more limited due to ambiguities in defining particle

size (Um et al., 2015) and issues such as coincident detection of multiple particles (Johnson et al., 2014; Lance, 2012). Stereoscopic and 2D imaging probes require fewer assumptions than scattering probes for the retrieval of microphysical information and have been deployed on both aircraft and weather balloons (Ulanowski et al., 2014).

  Holographic imaging instruments are effective at measuring a wide range of particle diameters, typically from a few microns up to a few millimetres (Ramelli et al., 2020). The sampling volumes of holographic instruments can be significantly larger

than for stereoscopic and 2D imagers, which reduces the number of required assumptions regarding statistical stationarity in the spatial distribution of cloud particles (Beals et al., 2015).

  Holographic imagers deployed on the ground (Henneberger et al., 2013; Schlenczek et al., 2017) or on cable cars (Beck et al., 2017) allow long-term cloud observations, though these are limited to mountain locations and measurements can be influenced by surface conditions (Beck et al., 2018). Aircraft-mounted holographic imagers allow targeted studies throughout the cloud

volume (Desai et al., 2019), but are limited by the high costs. A further limitation comes from the large forward velocity of the aircraft which can lead to problems with shattering of cloud ice particles unless specially designed anti-shattering probe tips are used (Korolev and Isaac, 2005). A tethered balloon allows long-term holographic observations at a fixed location (Ramelli et al., 2020), though such deployment is limited to favourable wind conditions and altitude ranges.

  Untethered sounding balloons can be deployed in most atmospheric conditions, allowing in situ measurements throughout

the full vertical extent of clouds. Various instruments have been deployed on sounding balloons for the study of clouds, including impaction sensors (Magee et al., 2020), cloud condensation nuclei counters (Delene and Deshler, 2001), 2D video microscopes (Takahashi et al., 2019), and standard radiosondes that measure pressure, temperature, and humidity.

  Remote sensing, with lidars, radars, or radiometers, can of course provide valuable information about clouds, and has the significant advantage of being able to provide better temporal resolution or coverage than sensors on aircraft, or balloons.

However, these techniques each have their own limitations; lidar can be limited by attenuation (Hogan et al., 2003; Mace and Protat, 2018) and multiple scattering of the probing beam (Weitkamp, 2005), radar measurements are strongly sensitive to the particle diameter which can complicate interpretations (Westbrook et al., 2010; Sassen et al., 2002), and radiometers can be unreliable for the detection of optically thick clouds and multi-layered cloud systems (Kuma et al., 2020). Similarly, satellite based remote sensing offers wide geographical coverage. In situ instruments have a role in the calibration and validation of the

remote sensing instruments (Morrison et al., 2020; Yang et al., 2018).

  Deploying instruments on aircraft requires a significant investment in engineering, as do many other sensors due to their relative complexity. The resultant cost precludes routine use of these instruments on sounding balloons, as is done with radiosondes and ozonesondes, because recovery of the instrument is desirable even if there is a telemetry link for the transmission of data. The need for recovery introduces sampling biases, especially at coastal sites where favourable wind conditions must be se-



lected. This is the primary motivation for the work presented here, where the advantages of the holographic approach are embodied in a low cost instrument. At this stage we have not implemented a telemetry link and instrument recovery is still necessary, but we note that this problem has been solved in the context of balloon-borne video microscopes (Murakami and Matsuo, 1990).

In this paper, manual analysis of the holographic dataset was performed. This manually analysed dataset was also used to
assess the performance of an automated analysis technique, but such a comparison goes beyond the scope of this paper and is left for a separate/companion paper. First we describe the actual instrument and choices made to ensure that it is sufficiently light, and reasonably low-cost so that the risk of non-recovery from a free balloon flight is acceptable. A summary of the flight is then given along with a description of the accompanying instruments, and then the results from that flight are presented.

In situ cloud measurements with which we can compare these observations in the South Australian region are limited,
though there have been several campaigns studying cloud over the nearby southern ocean (Boers et al., 1996, 1998; Wofsy, 2011; Ahn et al., 2017; McFarquhar et al., 2020). Air masses originating from the southern ocean region contribute significantly to South Australian weather systems (Chubb et al., 2011). In situ aircraft observation campaigns within southern ocean clouds have revealed that the cloud droplet number density is strongly seasonal, with significantly lower values measured during winter (Boers et al., 1998; Mccoy et al., 2020).

Wintertime flights have been undertaken north-west of Tasmania during the SOCEX-I experiment in 1993 (Boers et al., 1996), and south of Tasmania, as part of the HIPPO campaign (Wofsy, 2011), and more recently in an intensive campaign between 2013 - 2015 that we denote as "A17" in the later discussion (Ahn et al., 2017). A major aircraft campaign, the Southern Ocean Cloud Radiation Aerosol Transport Experimental Study (SOCRATES), was most recently undertaken in this region in 2018 (McFarquhar et al., 2020). However, flights were undertaken in summertime and are not as directly comparable
with results from this wintertime balloon launch.

## 2   The Instrument

The instrument is an imager that uses digital in-line holography (Schnars and Jüptner, 1994). A laser pulse is collimated and then directed at the sensor of a digital camera. Cloud particles in the sampling volume scatter light which interferes on the sensor with unscattered light (the reference beam) and the interference pattern that is recorded (the hologram) encodes information
about the nature of the scattering particles. That information can be extracted by analysis of the hologram to produce images at different planes in the sampling volume (Garcia-Sucerquia et al., 2006), using knowledge of the reference beam, which to a good approximation is a plane wave.

The camera is a See3Cam_CU51 (E-con Systems) which has a 5 MP sensor with dimensions of 5.70 mm x 4.28 mm. It was chosen in part for its low cost but more especially because it could be operated with a global shutter readout, which is important
because of the short laser pulse length of approximately 100 ns. The system is controlled by a Raspberry Pi computer, with an ancillary Arduino microcontroller that handles the laser pulse and camera readout timing.





The pulse width of the laser is approximately 100 ns, which is chosen as a compromise between the need to freeze the motion of micron-sized particles moving at up to 1 ms$^{-1}$, and having sufficient pulse energy (nominally 4 nJ) to properly expose the camera sensor. We use a diode laser with a wavelength of 405 nm and continuous wave power of 40 mW.

The laser is collimated in an attempt to make the magnification independent of the distance between the scattering particle and the sensor, which simplifies the analysis somewhat and extends the practically achievable sampling volume. The collimation attained was not perfect, yet the residual magnification effect was deemed acceptably small for this application. The impact of this residual magnification on particle diameter retrieval was corrected for using a numerical model and a resolution calibration target, following the procedure described in Chambers (2022).

There is a limit in how long (along the axis between the laser and camera) the sampling volume can practically extend; the angular subtense of the camera as seen by the scattering particle limits the scattering angle, or diffraction order, that can be captured, which in turn limits the resolution achievable (Garcia-Sucerquia et al., 2006). Thus the resolution for particle images closer to the laser is less than for those nearer to the camera. In our instrument we chose a sampling volume length of 70 mm which, because of a sun shield for the sensor, started about 50 mm from the sensor.

The collimated beam covered about half of the sensor, and this beam overlap multiplied by 70 mm defined the sampling volume. The diffraction-limited transverse resolution was around 5 microns at the camera end of the volume, and at the laser end the resolution decreased to around 12 microns, consistent with calibration measurements in the laboratory.

Figure 1 shows photographs of the instrument with and without the polystyrene foam insulation and the foil outer coating. The laser is housed in the smaller blue and white plastic enclosure seen behind the aluminium plate at the end of carbon fibre

tubes. Below the camera on the wider blue plastic structure are mounted the computer, batteries and Arduino microprocessor. The physical dimensions of the instrument are indicated on the figure, and the total weight of the instrument was around 1.5 kg. The majority of the weight came from the aluminium mounting plate which could be significantly reduced in future.

The sun shielding of the camera is primarily achieved by a bandpass optical filter, but also by setting the camera back from the aperture on the main box through which the laser beam enters. In addition, the window in front of the camera is a neutral

density filter with an optical density of 0.9. As discussed below, this was partially successful.

## 3   Balloon Flight

The holographic imager was launched at 0155 UTC on the 8th of August 2020 from 34.03° S, 138.69° E (north of Adelaide, South Australia), at an elevation of approximately 300 m. The launch location and balloon path are shown in Fig. 2.

Along with the holographic imager, the other instruments used in the flight were:

– a radiosonde (Vaisala RS41) for GPS tracking, Relative Humidity (RH) and air temperature measurements

– a camera (Raspicam with Raspberry Pi) for visual verification of cloud

– a polarsonde (Hamilton et al., 2020) to provide polarimetric optical backscatter measurements

– a datalogger recording measurements from a Bosch BME280 and a thermocouple (RH, pressure and air temperature),



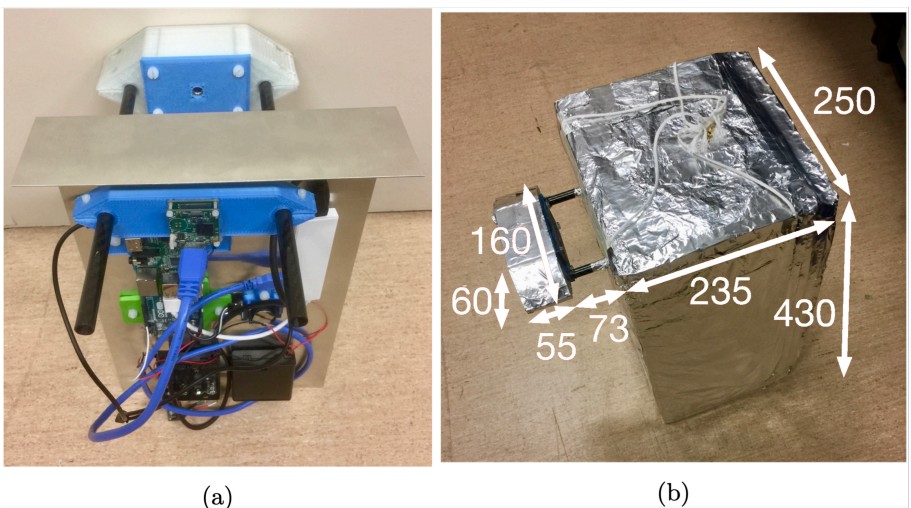

**Figure 1.** a) Core assembly of the holographic instrument showing the aluminium mounting plate, control electronics, carbon fibre spacing rods, and 3D printed laser mount. b) Final payload, with insulated housing, before launch with physical measurements overlaid in units of millimetres.

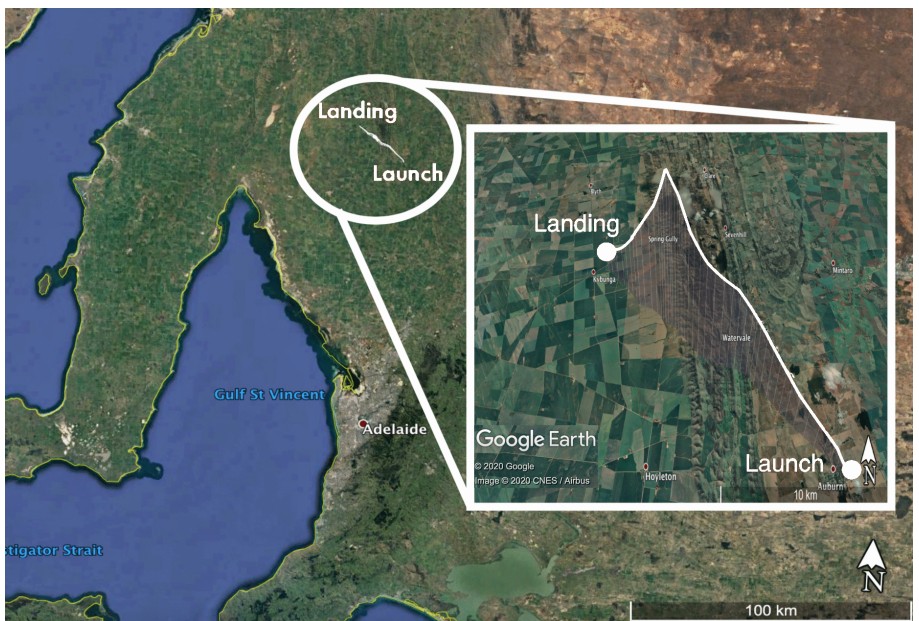

**Figure 2.** Map of the launch location and balloon path along with the surrounding region. Images obtained from © Google Earth, and CNES/Air-bus 2020.

and more detail on these is provided below.



The average ascent rate was approximately $4\,\mathrm{m\,s^{-1}}$ which was achieved using a 500 g balloon with approximately 3 m$^3$ of helium (at STP). The balloon reached an altitude of approximately 8.5 km after 30 minutes, travelling in a NE direction, and the payload train with its parachute was remotely cut down and retrieved, through use of the GPS tracking unit, approximately 20 km north-west of the launch site.

    To reduce the risk of losing the instrument in the sea to the west, balloon path forecasting incorporating Global Forecast
System (GFS) wind predictions was undertaken to select a launch day with suitable winds. In the event, the low-level winds were light and from the SE whereas at higher levels (above 8000 m) they were from the SW. The flight was terminated at an altitude of 8000 m because the camera indicated that the cloud top had been passed and the weather forecast model indicated that the balloon was about to enter the strong SW wind, which would have made recovery more difficult.

    An unbroken low-level stratus cloud was observed over the entire sky for the duration of the launch. Light rain and snow
were forecast by the Australian Bureau of Meteorology (BoM) for the nearby town of Clare in the morning and the launch was carried out during a time of no precipitation. The daily total rainfall measured by the BoM was 5.6 mm for Clare on that day, and light drizzle was observed at the launch site before the balloon was launched.

    A remotely operated mechanism was used to cut the payload train and parachute from the balloon. No damage to the instruments was identified on landing; however, the instruments began to tumble after being cut from the balloon which appears
to have dislodged the camera trigger cable, so that holograms were not obtained for the descent.

    The holographic instrument recorded one hologram per second. These data were written to an on-board SD card, rather than transmitting them to the ground via a telemetry link. This choice was made in the interest of instrument simplicity and to save weight, but adding this telemetry channel is an obvious future development as that would remove the imperative for payload recovery after a flight, which in turn removes biases introduced by a narrow selection of meteorological conditions.

This instrument did not suffer any sunlight saturation of the camera sensor at ground level; however, as the balloon ascended through the clouds, more indirect sunlight reached the sensor resulting in a significant fraction of the camera pixels becoming saturated. Inspection of the raw holograms revealed that the saturated pixels began at the bottom of the sensor and progressively higher rows of pixels became saturated as the balloon ascended. A correction to the sampling volume was made by subtracting the saturated pixels from the total number of pixels that lie within the spatial extent of the laser beam on the sensor. This
effective sampling volume was used to calculate the particle number density.

    The variation of sampling volume with altitude is shown in Fig. 3 and it is seen that the maximum sampling volume at ground was approximately 1.2 cm$^3$ and by the maximum altitude this had reduced to only around 0.2 cm$^3$. Though large ice crystals were observed at the highest altitudes, due to the reduced sampling volume and inherently lower number densities for ice crystals, a statistically significant number density measurement could not be obtained at these altitudes.

**4   Summary of Accompanying Instrumentation**

The position of the balloon was monitored in real-time from the ground using the GPS receiver in the Vaisala RS41 radiosonde. The radiosonde also provided air temperature and humidity measurements, but because this radiosonde was being used a second




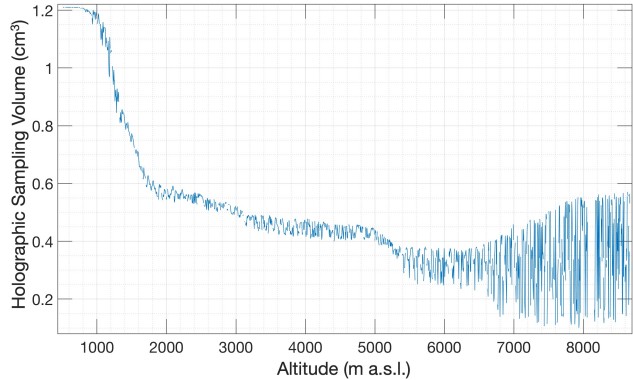

**Figure 3.** Variation of the holographic sampling volume with altitude due to sunlight saturation.

time we had doubts as to the reliability of the humidity sensor. Thus an additional data logger, described below, provided independent measurements of humidity. The RS41 uses a platinum resistance thermometer to measure temperature with a

reported resolution of 0.01 °C and a thin-film capacitor is used to measure relative humidity with a reported resolution of 0.1 % (Vaisala). Both sensors were sampled at 1-second intervals.

The Raspberry Pi camera was used to monitor the large-scale cloud conditions during the launch. The 6-megapixel colour images were recorded at approximately 30-second intervals, providing an independent test of whether the balloon was within cloud.

The polarsonde is a polarimetric backscatter instrument and is sensitive to the shape of the backscattering particles, which can be either aerosols or cloud particles. It is the same as that described in Hamilton et al. (2020), except that we used only the channel with the emitted light perpendicular to the scattering plane. The prelaunch procedure was also the same, though this time we mounted the instrument facing downward to mitigate the issue of sunlight saturation that was encountered above the cloud top (Hamilton et al., 2020).

The polarsonde was placed at the end of the payload train, with the holographic imager just one metre above, to simplify the comparison of the data from the two instruments. These instruments were suspended at a distance of approximately 9 metres from the balloon to reduce the impact of the balloon on the air sampled by the microphysics instruments. The camera and radiosonde were approximately 5 m and 6 m, respectively, above the holographic imager.

The data logger (made by Monash University) was attached to the polarsonde package at the bottom of the payload train. It

had a Bosch BME280 to measure pressure and relative humidity, and a thermocouple to measure air temperature. These were sampled at 1-second intervals. The thermocouple had a time constants of about 3 s.

### 4.1 Meteorological Measurements

The temperature and relative humidity measurements from the RS41 and Data Logger are plotted in Fig. 4. The data logger records systematically higher temperatures in the ascent, suggesting an uncalibrated offset in this sensor, likely compounded



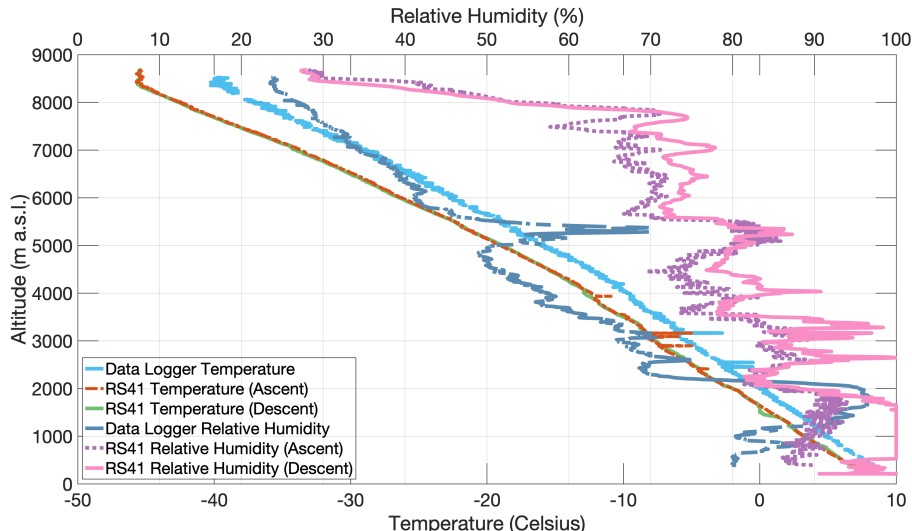

**Figure 4.** Temperature and relative humidity profiles from the RS41 radiosonde and Monash data logger (ascent only for the datalogger).

by the relatively long thermocouple time constant. The temperature values quoted herein are obtained from the RS41, which we judged to be the more reliable .

    The RH measurements of the RS41 and Data Logger show qualitative agreement for the large-scale features seen, such as the increase in RH up to approximately 2 km during ascent, as well as the peak in RH at around 5 km. These peaks were consistent with when the camera indicated that the instruments were in cloud.

The RS41 temperature measurements indicate that the melting level is at an altitude of approximately 1.9 km and the beginning of the tropopause is possibly seen at around 8.5 km, at the highest altitude of the flight. The RS41 made measurements reliably for the full launch duration, whereas the data logger sensors stopped working after reaching -40 °C, likely due to a drop in the battery voltage. The RS41 temperature profiles recorded during the ascent and descent agree well at each altitude to within a few degrees, suggesting that the meteorological conditions were stable for the flight duration and uniform horizontally

on a scale of 20 km, the distance between launch and landing locations.

## 5   Analysis of Holograms

The holographic dataset for the launch was manually analysed as follows; for each hologram, a 3D image (i.e. a stack of 2D images) was reconstructed using the Angular Spectrum diffraction method (Ratcliffe, 1956), the depths at which particles came into focus were identified, and on the resulting 2D images polygon masks were hand traced around the particle outlines. From

the masks we extracted the particle equivalent diameters, where the equivalent diameter of a particle is defined as the diameter of a circle with the same area as the drawn polygon.





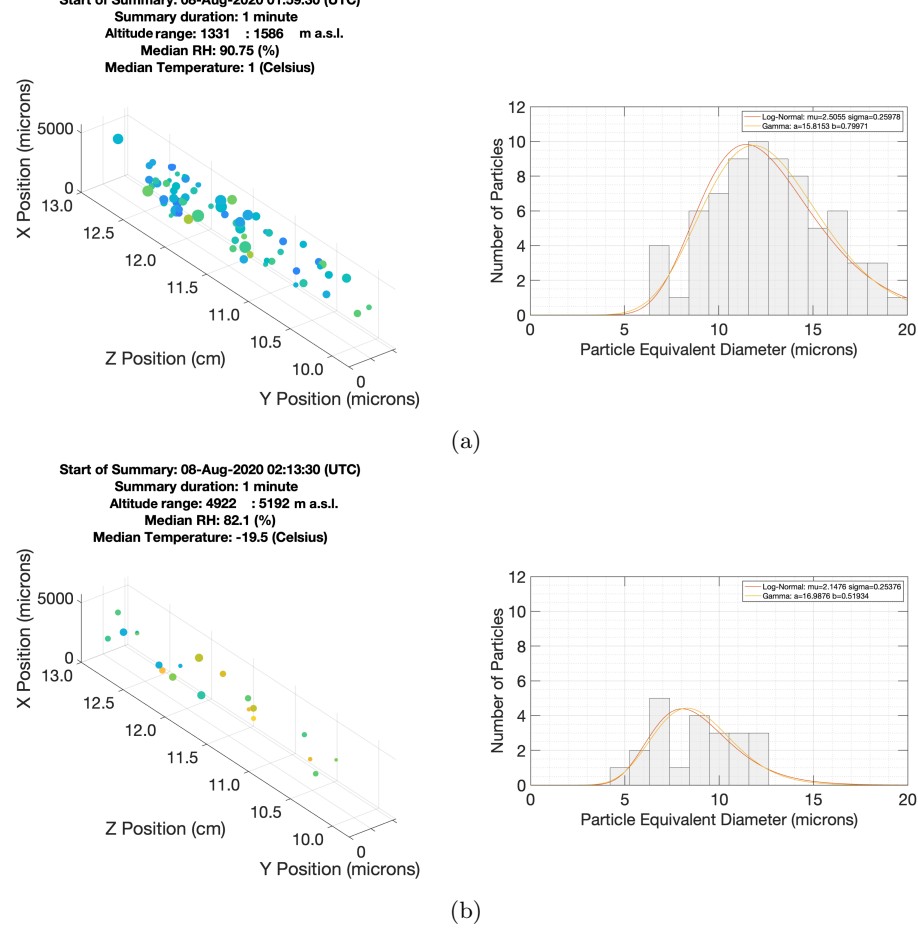

(a)

(b)

**Figure 5.** Summary of manually analysed holographic observations during the flight for: (a) the lowest identified cloud layer and (b) a higher altitude cloud layer. Particle 3D positions, size distributions, and median meteorological measurements are displayed. The z-axes on the 3D plots indicate the particle depth within the sampling volume relative to the camera sensor, and the transverse dimensions are in the plane of the camera sensor. Spheres on the 3D plots indicate the relative sizes of particles, but note that the absolute sizes are scaled for visibility.

An example of manually analysed holographic cloud observations is presented in Fig. 5 for two representative 1-minute time intervals (corresponding to vertical ranges of around 250 m each). Figure 5a corresponds to when the balloon was within the lowest observed cloud layer and Fig. 5b is for a higher altitude cloud layer. The 1-second sampling interval of the holographic
instrument allows for statistically significant distributions of particles to be recorded over these 1-minute averaging windows. The spatial positions of particles detected over these times appear to be uniformly distributed throughout the sampling volume, as seen in the left-most plots.

The measured particle size distributions during these times are shown in the right-most plots of Fig. 5 and the individual particle sizes are visualised by the relative sizes of the spheres that indicate the particle positions within the sampling volume.





Lognormal and gamma functions are overlaid in red and yellow, respectively, and fit the data well, as expected for typical clouds (Pruppacher and Klett, 2010).

## 5.1 Cloud Layers and their Properties

The most direct method to identify the cloud layers that the balloon passed through, provided the detected particle number density was sufficiently high, was to simply note the altitude at which the first, and last, cloud particles were detected by the

holographic imager, and define these altitudes as the cloud base and top respectively. This approach was validated by visual inspection of the Raspberry Pi camera images, and by the relative variations in RH.

Absolute number density measurements derived from the raw 1-second sampled observations are less reliable due to the relatively small numbers of detected particles. To compensate for this, in the following discussion we consider the 30-second averaged number density measurements which correspond to a spatial averaging range of approximately 120 m.

The full vertical profiles of holographic measurements of 30-second averaged particle number density and equivalent diameter are presented in Fig. 6. The pink shaded regions on the plot indicate altitudes at which the view of the Raspberry Pi camera was fully obscured by cloud and the RH is included for comparison. For clouds below 4000 m there is good agreement between each method as to the extent of the cloud bands. The slight offset between the shaded region and other cloud-detection metrics for the cloud bands around 3000 m is due to the coarse altitude resolution of the Raspberry Pi camera images. These

images are recorded once every 30 seconds, as opposed to once per second for observations from the other instruments.

Five distinct cloud bands were identified during the ascent, though two are fairly close and are grouped into band 2. Each of these cloud bands will be discussed in detail in the following sections. The in situ measurements for each of the identified cloud bands are summarised in Table 2 below (along with satellite-derived values for comparison).

## 5.2 Cloud Band 1

A sudden onset of particles at an altitude of around 600 m and a sharp termination in particle observations at an altitude of approximately 1900 m defines the extent of the lowest cloud band. From the ground this appeared as an unbroken low-level stratus cloud covering the entire sky. The Raspberry Pi camera images were fully obscured by cloud between these altitudes and the RH dropped significantly at the boundary of the cloud top, which correlates well with the holographic determination of cloud extent. The temperature decreased steadily from around 5 °C at the cloud base to -1 °C at the cloud band top.

Figure 7 shows holographic images from this cloud band, indicating that only spherical water droplets were present. An increase with altitude in the mean particle diameter is observed from around 9 microns to 13 microns between cloud base and top, as seen in Fig. 6. Light drizzle was observed before but not during the flight. The earlier drizzle would tend to remove the largest particles from the cloud and is consistent with the observation of only a few particles larger than 20 microns in diameter.

The number density, shown in Fig. 6, is first observed to increase and then decrease over an altitude range of around 500 m

from cloud base. It then increases again to an overall maximum at around 1700 m before decreasing rapidly towards the cloud top at around 1800 m.



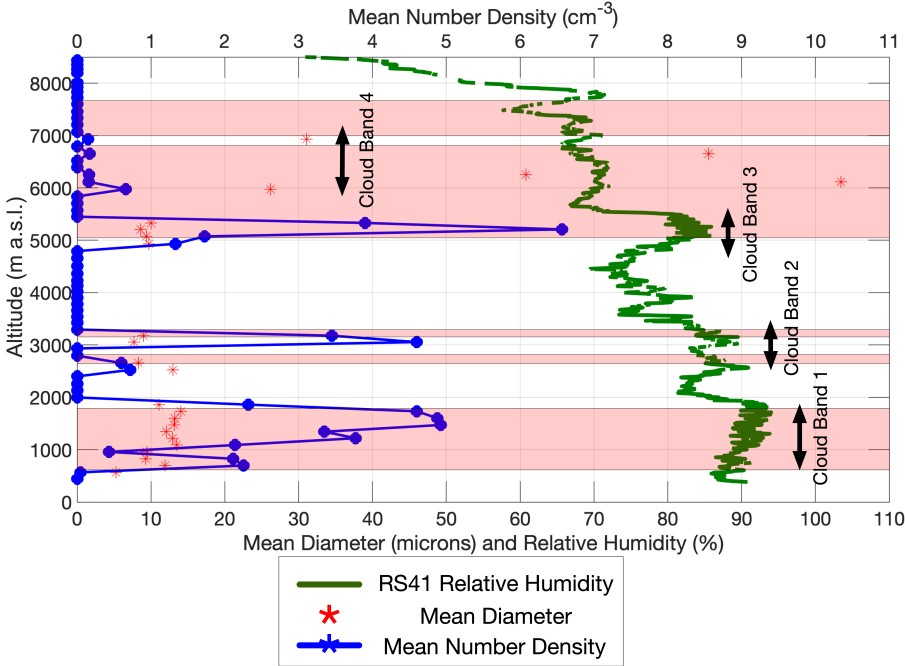

**Figure 6.** Full vertical profile of the 30-second averaged particle number densities and diameters measured by the holographic instrument. The pink shading indicates regions for which the Raspberry Pi camera images were determined to be fully clouded, and arrows denote the approximate bounds of the cloud bands determined by the holographic method. Relative humidity measurements from the RS41 radiosonde are plotted for comparison.

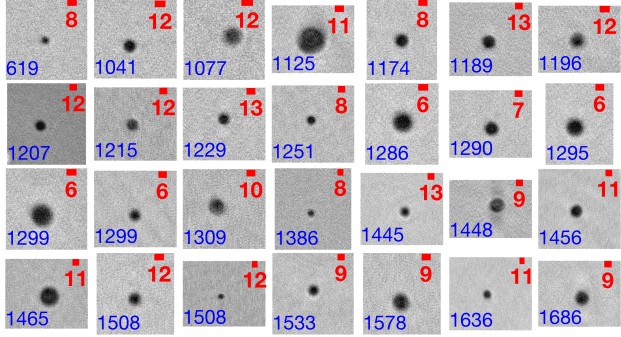

**Figure 7.** Representative particle images in the first band of cloud ranging from around 620 m to 1870 m. Altitudes of the detected particles are shown in the bottom-left corner of each image in units of metres. Scalebar units are in microns and note the differing scale for each image due to depth-dependent optical magnification.



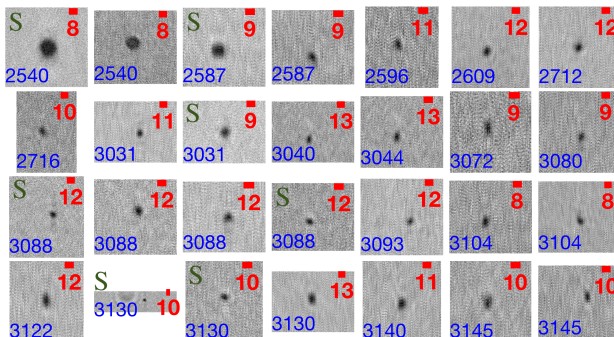

**Figure 8.** Representative particle images in the middle patchy bands of cloud ranging from around 2540 m to 3150 m. Altitudes of the detected particles are shown in the bottom-left corner of each image in units of metres. Scalebar units are in microns and note the differing scale for each image due to depth-dependent optical magnification. Particles considered more likely to have a symmetric shape are indicated by a green S.

## 5.3 Cloud Band 2

No particles were detected in the altitude range from the top of the first cloud band up to around 2500 m. In this altitude range the RH steadily decreases and then remains constant with altitude for about 300 m. The horizon is visible in the Raspberry Pi camera images during this interval, supporting the assertion that the balloon was between cloud layers.

The next detection of particles at around 2500 m coincided with a peak in the RH and the camera images again became fully obscured by clouds. Two distinct cloud bands (2a and 2b) were identified between this altitude and about 3200 m, with a vertical separation between the bands of only around 300 m. However, the particle images were so similar that these are grouped into band 2. Representative particle images from both cloud bands are shown in Fig. 8.

Visual inspection of the cloud particle images indicates that this cloud may consist of a mixture of irregularly shaped ice crystals with a mean effective diameter of 9 microns, as well as particles that appear circular that have been labelled with an S on Fig. 8, suggesting that this was a mixed phase ice/liquid cloud. This interpretation is presented with a lower confidence as the particles are at the lower resolution limit of the holographic instrument and specific ice particle habits cannot be identified. The temperature within the cloud ranges from approximately -5 °C at cloud base to -8 °C at cloud layer top, which is suitable for the formation of ice crystals.

The vertical profile of cloud particle number density, as seen in Fig. 6, shows that cloud layers 2a and 2b are no greater than 200 m in thickness, which limits the investigation into number density variability (only about 50 holograms are obtained in 200 m of ascent). However, it is interesting to note that the number density measured in the layer 2a is significantly lower than that in layer 2b. The RH peaks within each of the cloud masses and sharply drops within regions of clear air.



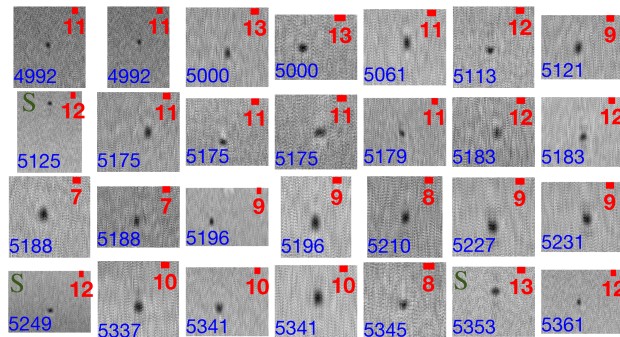

**Figure 9.** Representative particle images in the middle patchy bands of cloud ranging from around 4990 m to 5380 m. Altitudes of the detected particles are shown in the bottom-left corner of each image in units of metres. Scalebar units are in microns and note the differing scale for each image due to depth-dependent optical magnification.Particles considered more likely to have a symmetric shape are indicated by a green S.

## 5.4 Cloud Band 3

The vertical extents for clouds identified above 5000 m by the camera and the holographic imager do not agree as well as for the lower clouds; these clouds were less optically dense in the Raspberry Pi camera images. The cloud extents determined by the holographic imager tended to agree better with the measured variations in RH.

A band of cloud was identified by the holographic detection of particles, summarised in Fig. 9, and by inspection of the Raspberry Pi camera images. This band began at an altitude of approximately 4990 m and the last detected particle was at an altitude of around 5380 m. This corresponds to a cloud thickness of approximately 390 m.

As with each of the lower cloud bands, a sharp rise and then a fall in RH is well correlated with the detection of cloud particles. Whilst the Raspberry Pi camera images remain obscured by cloud throughout this altitude range, a subtle thinning is noted in one of the frames that coincides with the drop in detected particles at an altitude of around 5380 m.

The temperature decreased from -19 °C to -22 °C within cloud band 3, and the particle sizes are similar to those seen in cloud band 2. The similarity in particle properties to the significantly lower-in-altitude cloud band 2 is noteworthy. The number density profile for this band is shown in Fig. 6, though the effective sampling volume had reduced to only 0.4 cm$^3$ at this altitude.

## 5.5 Cloud Band 4

Only five particles were detected above around 6000 m which were seen to be large ice crystals with complicated shapes, as shown in Fig. 10. Given the reduced sampling volume at these altitudes due to sun saturation, along with the lower number density of ice crystals, the number density measurements from the holographic instrument in this altitude range are not statistically significant.



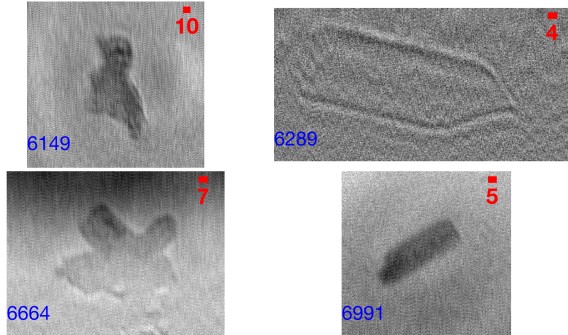

**Figure 10.** Particle images in the thin top band of cloud ranging from around 6010 m to 6990 m. Altitudes of the detected particles are shown in the bottom-left corner of each image in units of metres. Scale bar units are in microns and note the differing scale for each image due to depth-dependent optical magnification.

Due to the small number of detected particles in this band of cloud, the holographic method of determining cloud extent was less reliable than the Raspberry Pi camera method. The first particle was identified at an altitude of around 6010 m and the Raspberry Pi camera images indicate that the cloud becomes more transparent with altitude, allowing more sunlight to reach the camera sensors, up to an altitude of around 6500 m. Above this altitude the optical thickness of the cloud becomes significantly more variable, with patches of embedded clear air also noted in the images. The highest particle was detected at an altitude of 6990 m and the camera images indicated that the balloon had fully exited the cloud at an altitude of around 7800 m.

Despite the proximity to cloud layer 3, this band of cloud exhibits distinctly different microphysics. This contrast is primarily noted from the particle images, as shown in Fig. 10. No small ice crystals were detected, and the mean particle equivalent diameter increased by an order of magnitude from 8 microns to 61 microns. This was the first time such large ice crystals were detected during the flight.

## 6 Comparison with Previous Observations

We can compare our measurements to wintertime flights undertaken north-west of Tasmania during the SOCEX-I experiment in 1993 (Boers et al., 1996), and more recently in a campaign between 2013 - 2015 undertaken to the south-west of Tasmania that we denote "A17" (Ahn et al., 2017). The particle number densities and particle effective diameters for these campaigns are shown with our measurements in Table 1. The particle size distributions obtained within each stratus cloud band detected in the balloon launch are summarised in Fig. 11.

The measurements made in the SOCEX-I flights concentrated on stratocumulus cloud layers that were centred at pressures (altitudes) between 950 hPa and 850 hPa. They are thus most readily comparable to cloud layer 1 from our balloon flight. The numbers from SOCEX-I that we quote in Table 1 are the medians of the diameters and number densities within the cloud layer studied in each of the five reported flights, expressed as a range, to compare more directly with our measurements. In





| Layer | Diameter ($\mu$m) | | | Number Density (cm$^{-3}$) | | |
|---|---|---|---|---|---|---|
| | Holographic | SOCEX-I | A17 | Holographic | SOCEX-I | A17 |
| | | $16 - 44$ | 22.8(6.0) | | $15 - 100$ | 40(41) |
| 1 | 13(4) | | | 4(2) | | |
| 2a | 9(3) | | | 4(1) | | |
| 2b | 8(1) | | | 7(3) | | |
| 3 | 9(2) | | | 6(3) | | |
| 4 | 61(34) | | | | | |

**Table 1.** Comparison between cloud properties measured in this balloon flight, and in the SOCEX-I and A17 campaigns. For the holographic measurements, the means and standard deviations of equivalent diameter and number density are specified for each cloud band.

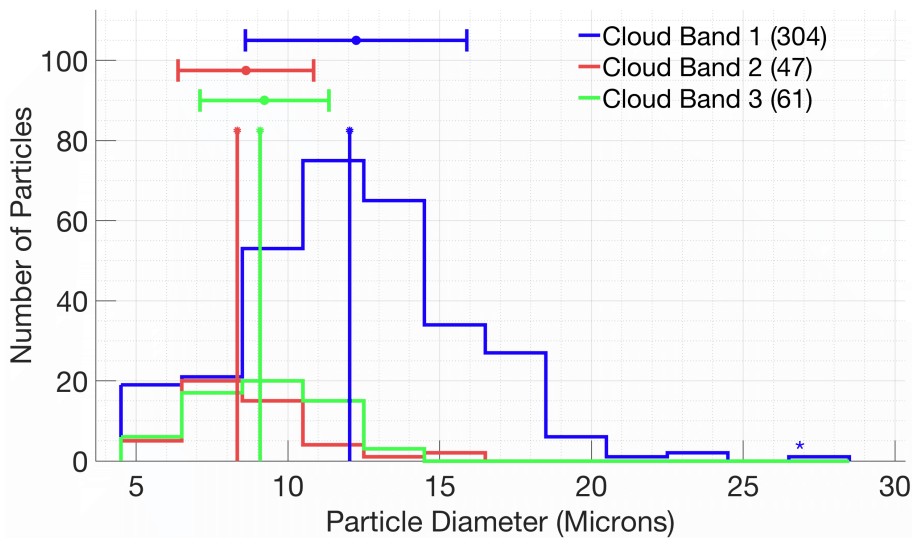

**Figure 11.** Particle size distributions for particles within each band of stratus cloud detected in the balloon launch. Error bars indicate the mean and standard deviations of each histogram, and the stem plots indicate the median values. The number of particles contributing to each histogram is shown in the legend. The star above the far-right column signifies that this column includes contributions from all data points with value larger than or equal to the bin range. The bin width is 2 microns.

Boers et al. (1996) averages over full cloud bands are not reported, rather they show averages over 10 hPa pressure intervals. Significant discrepancies were noted in Boers et al. (1996) between particle diameter observations from the two cloud probes, yet we follow their methodology in considering the inclusion of observations from both instruments to be the most reliable approach.

     The A17 campaign consisted of 20 flights south-west of Tasmania under a range of synoptic conditions. Cloud layers were 295   centred between 650 hPa and 920 hPa, which make these observations again most comparable with cloud layer 1 from our



balloon flight. For liquid-only clouds, both the reported average number density and the average effective diameter depended only a little on the averaging methodology, however the effective diameter depended strongly on the selection of measuring instruments – see Table 2 of Ahn et al. (2017). We are comparing our measurements in layer 1 with the "consistent liquid average" (row 5) from that table.

Compared to the two aircraft measurement campaigns the number densities measured in the balloon launch are low and could be taken as a sign that the holographic instrument is failing to detect some particles, since we are operating close to the resolution limit of the instrument. However, clouds with number densities below 10 cm$^{-3}$ and comparable particle diameters have been measured in the southern ocean (Mccoy et al., 2021), and other parts of the world (Wood et al., 2018; O et al., 2018). A recent review of observations from the southern ocean (Mace et al., 2021) reveals that clouds with number densities as low 305 as those measured in this balloon launch occur with significant frequency over this region.

    It is therefore of interest to compare the balloon measurements with those from a general review of in situ measurements within stratus clouds (Miles et al., 2000). This comparison is displayed for mean particle diameter in Fig. 12a. The mean particle diameters measured in this launch are in the lower range for marine clouds and the upper range for continental clouds.

    These intermediate values may be as a result of the proximity of the launch site to both coastal and continental regions. 310 This interpretation is supported by HYSPLIT modelling which indicates contributions to the air mass from both continental sources and the southern ocean. A summary of the HYSPLIT modelling can be found in Fig. S1, which is available as part of the dataset described in the Data Availability section of this paper. The number density comparison is shown in Fig. 12b and it is again noted that the measurements from this launch are at the lower limit of those seen in previous campaigns.

## 7   Polarsonde Observations

The primary polarsonde observables are the polarised backscatter components that are co- and cross-polarised with respect to the emitted polarisation. The instrument is sensitive to backscatter from both cloud particles and aerosols, and so a key challenge is in separating the contributions of these populations of particles. In the conditions encountered here, with relatively low cloud particle concentrations, the aerosol contribution completely swamped the cloud contribution.

    The vertical profiles of the polarsonde backscatter components are shown with the cloud bands overlaid, in Fig. 13. It is 320 clear that the variation in the polarsonde signal channels is not correlated with the presence or absence of clouds, though some changes in the vertical gradient of the signals are located at cloud layer boundaries.

    Now in Hamilton et al. (2020), where the polarsonde was flown on balloon soundings from Macquarie Island, there was in one flight a signal (component) that was correlated with the presence of cloud and it was estimated in that case that the cloud particle number density was 40 cm$^{-3}$. In some subsequent balloon flights no signal correlated with cloud, that was nevertheless 325 visually apparent, and only an aerosol component was seen. The holographic imager in the sounding reported here measured cloud particle number densities an order of magnitude less than that for the successful polarsonde detection of cloud layers at Macquarie Island. Thus the failure of the polarsonde to detect cloud lends credence to the low value of particle number density derived from the holographic imager.





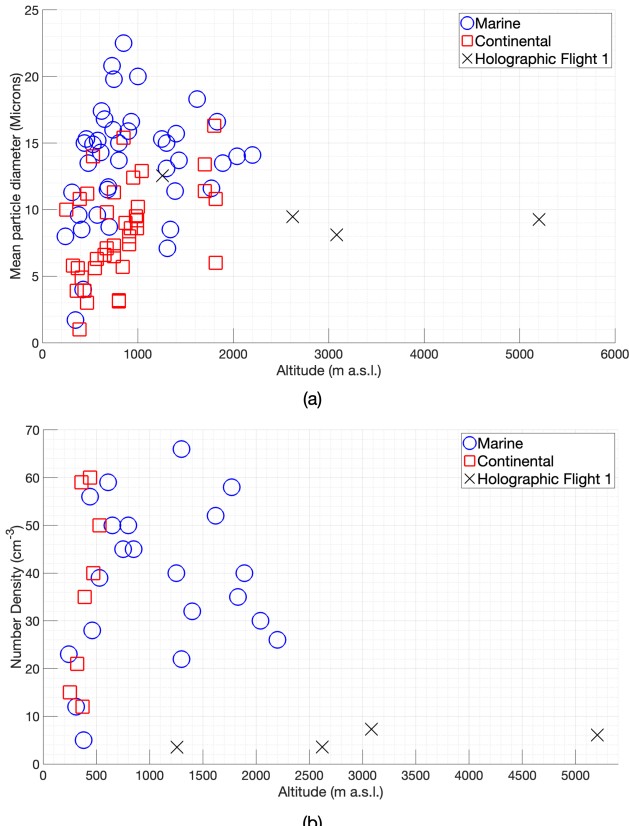

**Figure 12.** Mean particle diameters (a) and number densities (b) measured during this launch in cloud bands 1, 2, and 3, compared to those for stratocumulus clouds in the literature (Miles, 2000). Note that cloud band 2 has been split into the two constituent cloud masses, and cloud band 4 is removed from this comparison due to the small number of particle detections. Significantly larger number densities were reported within the Miles 2000 dataset, but only the smaller values are shown here for comparison.

## 8 HIMAWARI-8 Comparison

The in situ measurements from the holographic imager and RS41 radiosonde can be directly compared to HIMAWARI-8 (Bessho et al., 2016) satellite retrievals of Cloud Effective Radius (CER), Cloud Top Height (CTH), and Cloud Top Temperature (CTT) (Ishida and Nakajima, 2009; Kawamoto et al., 2001). HIMAWARI-8 data used in this study were obtained for 0230 UTC at which time the balloon was at approximately its maximum altitude. The satellite data products have a spatial resolution of 0.05 degrees in longitude and latitude. It is assumed in this study that the satellite-derived CER values are representative

of the highest altitude cloud layer within a pixel since thermal emission from the cloud predominantly comes from the cloud top (Hamann et al., 2014).

Significant spatial variation was seen in the HIMAWARI-8 Cloud Type retrieval in the region surrounding the launch site, which is consistent with the multi-layer cloud system detected during the launch. A map of the Cloud Type retrievals in the



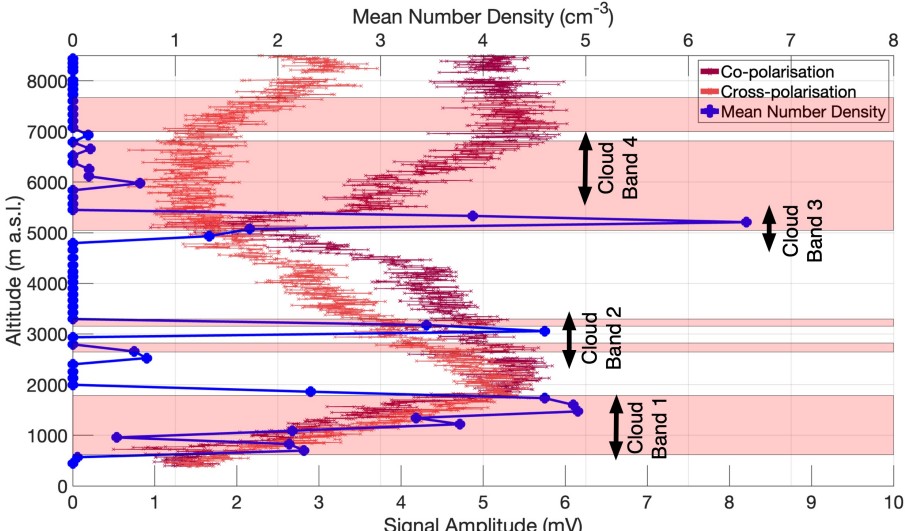

**Figure 13.** Vertical profiles of the polarsonde backscatter signals. These profiles are compared with the vertical profile of 30-second averaged holographic number density. Pink shaded regions indicate cloud bands identified by the Raspberry Pi camera.

surrounding region can be found in Fig. S2, which is available as part of the dataset described in the Data Availability section of
this paper. A large-scale stratus cloud layer is seen, along with more localised regions of nimbostratus and altostratus, as well as a thin cirrus layer adjacent to the launch site. Contiguous regions of each cloud type are identified in the area surrounding the launch site in the Cloud Type retrieval, and so it is assumed that spatially averaged properties of the cloudy pixels surrounding the launch site will be representative of the cloud layers directly above the launch site that are obscured by the highest altitude clouds in the satellite retrievals.

Pixels in the satellite retrievals within a 3 x 3 degree region surrounding the field site were averaged within three ranges, as defined by the CTH. Low-level clouds were defined as having a CTH smaller than 2 km, mid-level clouds had a CTH in the range from 2 km to 6 km, and high-level clouds had a CTH greater than 6 km. In this classification, cloud band 1 (in our balloon flight) is low-level, cloud bands 2 and 3 are mid-level, and cloud band 4 is high-level. These ranges were chosen to be consistent with the altitudes at which distinct cloud types were identified from the launch observations, and broadly correspond
to features in the histogram of satellite-retrieved CTH values.

The spatial region used to compute the averaged values is displayed in Fig. 14 which shows the retrieved CER. Small particle sizes are seen in the low-level and mid-level stratus clouds surrounding the launch site with larger values identified within the high-level cirrus cloud, qualitatively consistent with that measured by the holographic imager. Each of the four cloud bands identified during the launch, as summarised in Fig. 6, are compared here. Cloud band 2 has been separated into the two
constituent clouds bands to maximise the number of comparisons.

A summary of the comparison between in situ and HIMAWARI-8 measurements of CTH, cloud particle diameter, and CTT is presented in Table 2. The in situ balloon determination of CTH was based on the holographic method of identifying the



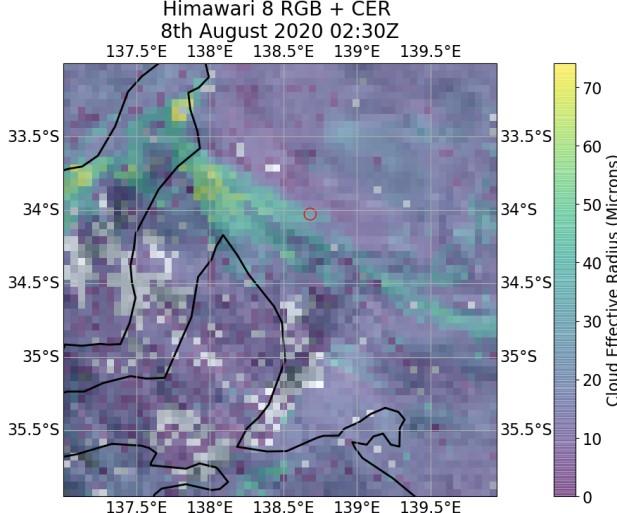

**Figure 14.** HIMAWARI-8 retrieval of Cloud Effective Radius (CER), 8 August 2020 02:30 UTC. The red circle indicates the location of the launch site.

altitude of the highest detected particle within a cloud band. The CTT is then determined from the temperature measured at this altitude by the RS41 radiosonde. The uncertainty with this method is difficult to quantify as it is dependent on the number density of particles, and the holographic imager is not sensitive to particles smaller than around 5 microns. The in situ measurements of CTH and CTT are rounded to a lower precision to attempt to account for these issues.

The CTH and CTT comparisons of greatest interest are for the lowest and highest cloud bands, since the mid-level range encompasses three distinct stratus layers of differing CTHs. For cloud band 1 the mean HIMAWARI-8 CTH is 340 m lower than that identified by the holographic imager and does not agree with the holographic value within one standard deviation from the mean. The HIMAWARI-8 CTT agrees with the RS41 value to within the uncertainty for this cloud band.

In cloud band 4 the HIMAWARI-8 CTH is again lower than that from the holographic imager, with a larger difference of 610 m in this case. The holographic value does not lie within two standard deviations of the HIMAWARI-8 value for this cloud band. The HIMAWARI-8 CTT is 5.3 °C larger than the RS41 measurement which does not lie within the range of HIMAWARI-8 measurements. For both measurements, the 95 % confidence intervals of the measured values do not overlap, suggesting that these differences may be significant.

The results of this study are consistent with a recent HIMAWARI-8 comparison (Huang et al., 2019) with Clouds, Aerosols, Precipitation, Radiation, and Atmospheric Composition over the southern ocean (CAPRICORN) shipborne observations (Mace and Protat, 2018) and the CALIPSO satellite (Winker et al., 2010) over the southern ocean, which found satellite retrievals of CTH to be significantly lower than the shipborne observations for warm liquid clouds, along with a tendency for the satellite measurements to misclassify lower-level clouds as cirrus.



| Cloud Label | Cloud Base Height (m) | Cloud Top Height (m) | | | Diameter (µm) | | | Cloud Top Temperature (°C) | | |
|---|---|---|---|---|---|---|---|---|---|---|
| | Holographic | Holographic | HIMAWARI | Diff. | Holographic | HIMAWARI | Diff. | RS41 | HIMAWARI | Diff. |
| 1 | 620(10) | 1870(10) | 1534(307) | −340 | 13(4) | 26(8) | 13 | −1.3(0.1) | −1.5(1.8) | −0.2 |
| 2a | 2540(10) | 2720(10) | 3849(1018) | 1130 | 9(3) | 32(11) | 23 | −6.3(0.1) | −12.5(5.5) | −6.2 |
| 2b | 3030(10) | 3150(10) | 3849(1018) | 700 | 8(1) | 32(11) | 24 | −8.6(0.1) | −12.5(5.5) | −3.9 |
| 3 | 4990(10) | 5380(10) | 3849(1018) | −1530 | 9(2) | 32(11) | 23 | −21.9(0.1) | −12.5(5.5) | 9.4 |
| 4 | 6010(10) | 6990(10) | 6383(270) | −610 | 61(34) | 52(32) | −9 | −33.6(0.1) | −28.3(2.1) | 5.3 |

**Table 2.** Comparison between cloud properties measured during the launch and by the HIMAWARI-8 satellite. Measurements from the in situ instruments are quoted with uncertainties specified within brackets. Means and standard deviations are specified for each satellite-retrieved cloud parameter within the CTH ranges defined earlier. Differences are specified as the in situ value subtracted from the HIMAWARI-8 value.





The HIMAWARI-8 retrievals of CED can be compared for all altitude ranges, due to the similarity in the particle size distributions for each of the mid-level cloud bands. At mid-level and low-level CTH ranges the HIMAWARI-8 retrievals are significantly larger than from the holographic imager. Measurements from both instruments have notably larger uncertainties for the high-altitude cirrus layer, and these measurements were found to agree within these larger uncertainties. Significant biases between HIMAWARI-8 CER retrievals and in situ measurements from SOCRATES aircraft flights over the southern ocean have previously been identified (Kang et al., 2021).

## 9    Conclusions

An untethered balloon launch of a holographic imager into clouds was reported here. Multiple bands of stratus and cirrus cloud were detected by the holographic instrument, as independently validated by the collocated RS41 radiosonde measurements, Raspberry Pi camera images, and polarsonde observations. The detected clouds were determined to be a low-level stratus cloud consisting solely of water droplets, multiple mid-level stratus clouds composed of small particles, and a high-altitude cirrus layer of large ice crystals with significantly more complicated particle morphologies.

Measured particle diameters for ice crystals and water droplets were compared with and found to be consistent with previous in situ measurements for these cloud types, with a focus on comparing with measurements from the surrounding southern ocean region from which a significant component of the observed air mass was believed to originate. The measured number densities were particularly small and whilst similar values have been measured previously, particularly in the southern ocean region, it is suggested that this may indicate a potential sampling bias. Future launches should be carried out alongside independent cloud sampling instruments to test this interpretation, but it is expected that such a limitation could be overcome in future by increasing the instrument sampling volume through the use of a larger camera sensor and by improved optical filtering to avoid sunlight saturation issues.

A secondary focus of the launch was to use collocated holographic measurements to assist with the interpretation of polarsonde observations. The backscatter signals were found to be dominated by scattering from aerosols, rather than cloud particles. Variations in the overarching trends were linked with the presence of cloud tops and bases, but otherwise no correlation was seen between the polarsonde signals and the presence or absence of cloud. This lack of correlation lends credence to the low number density measurements from the holographic imager.

A preliminary comparison between the in situ cloud observations and remote retrievals from the HIMAWARI-8 imaging satellite was presented. These early results suggest significant differences between in situ measurements and the satellite retrievals of Cloud Top Height, Cloud Effective Diameter, and Cloud Top Temperature, consistent with recent reports from the SOCRATES and CAPRICORN observation campaigns over the southern ocean. Routine launches of holographic instruments would allow a significant increase in the availability of cloud microphysical measurements, as required for robust calibration and validation of remote sensing methods, and for the improvement of climate and weather models.



*Data availability.* Data required to reproduce the key figures in this paper are available via Zenodo at: https://doi.org/10.5281/zenodo.10297799. Raw holograms are available on request.

*Author contributions.* TEC and MH designed the instrument and analysis tools, undertook the field work, and interpreted the results. MH and IMR supervised the work. TEC and MH prepared the manuscript and all authors reviewed the contents.

*Competing interests.* The authors declare that they have no conflict of interest.

*Acknowledgements.* The authors wish to acknowledge the logistical and technical support of Mark Jessop and the Amateur Radio Experimenters Group (AREG) with regards to the preparation, launch, and tracking of the balloon used in this work.

Research product of Cloud Property (produced from Himawari-8) that was used in this paper' was supplied by the P-Tree System, Japan Aerospace Exploration Agency (JAXA).

Financial support from the University of Adelaide, Atrad Pty Ltd, and the Institute for Photonics and Advanced Sensing (IPAS) is acknowledged by the authors.



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
