# Peer review of "A Light-Weight Holographic Imager for Cloud Microphysical Studies from an Untethered Balloon"

_EGUsphere, 2023_

## Author Comment (AC2)

**Responses to Referees:**

*Referee 1:*

This manuscript describes a low-cost holographic imager on a radiosonde, and shows that it works well, compared to other measurements. This is very important, because radiosondes often have very little information about cloud types and microphysical properties. Additionally, the manuscript is well written and well presented. I suggest publication as is.

**Response to Anonymous Referee 1:**

We thank the referee for their time and positive feedback.

**Response to Anonymous Referee 2:**

Below are the referee comments in black and our responses in blue. Specific edits within the manuscript are highlighted in yellow and are associated with their line number in the new manuscript. A pdf created using the latexdiff package is also provided highlighting changes between the current and previous manuscript versions.

**NB: We had some issues with the latexdiff package and so the line numbers may be inconsistent with the new manuscript, and certain changes (such as the merging of paragraphs by removal of line breaks as pertinent to Major Issue 1 discussed below) are not present in the latexdiff output. The line numbers quoted in this response are valid for the new manuscript and this should be consulted directly for the most reliable assessment of our implemented changes.**

*Referee 2:*

**Summary:** The authors describe the construction and testing of a radiosonde like holographic cloud droplet detector. While digital holographic detectors have been used previously in the laboratory, aircraft as well as tethered balloon or gondola measurements, this paper attempts to create a 'disposable' or 'radiosonde like' holographic detector for cloud droplet and ice particle detection. Such an instrument if successfully built will provide valuable measurements of cloud particle profiles along with regular radiosonde temperature and humidity measurements. The paper describes how the authors built the instrument and measurements from a test flight. The authors also compared their obtained cloud particle properties with other local measurements as well as satellite retrievals to showcase the accuracy of the untethered holographic instrument.

**Recommendation: Reconsider with major revisions or resubmission**

Overall, the context of the paper describing the new instrument design and test flight is very valuable to the atmospheric community. However, there are a few major issues with the data, the analysis and the writing of the manuscript. Due to the valuable nature of the instrument, I recommend acceptance but with major revisions. I have listed the major and minor issues below that need to be addressed by the authors before this manuscript can be considered to be published.

*Response:*

We thank the referee for their time and positive feedback. We have responded to all comments and believe that their suggestions have significantly improved this manuscript.

**Revisions:**

**Major Issue 1: The writing**

In order for this manuscript to be published, the writing needs to be significantly improved. As it currently stands, each couple of lines seem to be separated into paragraphs of their own. In a scientific article, each paragraph should represent a new idea or point. For e.g., Lines 16-20 seem to break the chain of the introduction and should be combined with Lines 59 onwards. The first paragraph (lines 11-14 can then be combined with the 3rd paragraph lines 20-27 to create one cohesive paragraph detailing the importance of clouds and cloud observations. Conjunctions such as 'Meanwhile' or 'Similarly' can help stitch together slight changes of topics but with the same underlying idea or principle.

*Response:*

We have made updates throughout the manuscript based on this comment to better capture new ideas/points inside their own longer paragraphs. We have implemented the specific comment about combining the first and third paragraphs and shifting the content between them to later in the section.

**Major Issue 2: The RH measurements**

Figures 4 & 6 show the RH values along with identified cloud bands and particle number densities. For a cloud band to exist, RH values should be close to 100% if not slightly larger. Yet cloud bands are being identified and cloud droplets and ice being detected at subsaturated conditions.

*Response:*

Relative humidity (RH) measurements with respect to liquid water within clouds observed in this launch ranged between approximately 70 % to 90 %. These values are somewhat low, though

we note that mechanisms exist for <100 % relative humidities within clouds, such as the entrainment of dry air within pockets inside the cloud eg. Korolev and Isaac (2006). We also note that similarly low RH values within cloud have been reported in previous in situ field experiments, see Schuyler et al. (2019) or Ramelli et al. (2020) for example. The lower values reported in the latter reference are attributed to challenges in measuring RH within clouds.

Furthermore, we direct the referee to the publicly available radiosonde sounding data obtained by the Australian Bureau of Meteorology (BoM) around 2 hours before our balloon launch (launched from Adelaide Airport, which is approximately 100 km to the south of our launch site) that is hosted online by the University of Wyoming at the following website:

https://weather.uwyo.edu/cgi-bin/sounding?region=pac&TYPE=TEXT%3ALIST&YEAR=2020&MONTH=08&FROM=0800&TO=0812&STNM=94672&ICE=1&REPLOT=1

We note that their calibrated radiosonde reports similarly low relative humidity (with respect to liquid) values for the altitudes corresponding to high altitude cloud bands identified in this launch, yet the relative humidity (with respect to ice) is seen to be around 100 % for these altitudes. We do not emphasise this level of detail in the manuscript since we are not drawing conclusions pertaining to the absolute values of the relative humidity measurements.

Our RH measurements were obtained from a previously deployed RS41 radiosonde and we are not able to calibrate this sensor with our current resources. We acknowledged this limitation in the original manuscript, but we do not believe it is an important one for this paper since we were only using these measurements as a qualitative and independent test for when the balloon had passed through cloud, based on the relative variations in RH. To make this point clearer, we have added the following paragraph, including reference to the independent BoM measurements, to the manuscript beginning at:

Line 176: "RH measurements with respect to…"

Looking at Figure 4, while local peaks in RH seem to coincide with the Raspberry Pi camera and holographic images, the mean RH value seems to have an offset along with a decreasing bias with altitude. Please check and recalibrate the RH values.

*Response:*

We assume the referee is actually referring to Figure 6 (where the comparison between each instrument is summarised) as opposed to Figure 4 (which shows only the meteorological measurements).

Regarding a potential offset;

In the manuscript we present three independent methods for determining when the balloon is within cloud:

1.Relative variations in the RH

2. Manual inspection of the outwards facing webcam photos (ie. Checking when it is obscured by cloud)

3. Identification of particles by the holographic imager.

We have checked the data again and see no evidence for an offset due to processing of the data. Since each method is sensitive to fundamentally different physical parameters, we do not expect perfect agreement. Additionally, we expect slight differences between the measurements due to their differing sampling rates. The holographic imager and RH observations are obtained at 1 second intervals, and the holographic measurements of number density and particle diameter are averaged to 30 second intervals. The Raspberry Pi camera reports images at approximately 40 second intervals. A 10 second interval corresponds to approximately 40 metres of vertical distance with an average ascent rate of around 4 m/s. We have updated the manuscript to give a more precise value for the Raspberry Pi camera sampling rate, and have added the following paragraph to clarify this matter beginning at:

Line 242: "For clouds below 4000 m…"

Regarding the decreasing bias with altitude;

As noted above, we are unable to calibrate the previously used RS41 radiosonde and so we acknowledge within the manuscript that this is a limitation of the study and we cannot rule out potential calibration issues. Even for calibrated radiosondes, we note that significant variability is seen in RH reported from different sensors with a strong dependence on temperature, as potentially seen in this launch, see for example Figure 8.3.5 of Nash et al. (2011). The calibrated radiosonde launched near to our field site by the Australian Bureau of Meteorology around 2 hours before our launch, referred to earlier in our response, shows a similar trend of decreasing RH with altitude.

We avoid potential absolute calibration issues by focussing only on the relative variations in relative humidity to independently denote the presence of cloud bands. We believe the approach of using relative variations in relative humidity as an independent indication of cloud layers is well motivated and note that it has been used in previous works eg. Schuyler et al. (2019). This matter is addressed by the following paragraph added to the manuscript beginning at:

Line 176: "RH measurements with respect to…"

**Major Issue 3: Band 3 measurements**

Figure 9 shows cloud particles detected by the holographic imager between 4990 and 5380 m altitude. Temperatures at these altitudes are correctly between -19 ∘C to -22 ∘C. Yet the particles look less like ice, but similar to particles from band 1 or 2. Are these images correctly labeled as band 3? A few particles also show blurring, possible due to a combination of camera

and particle motion. If the particle diameters are determined by hand tracing their shape, significant errors in particle size may occur.

*Response:*

We have checked the data again and confirm that the images are correctly labelled as band 3. Given that these particles are close in diameter to the resolution limit of the holographic imager, we do not wish to speculate as to why they appear similar to those in the previous cloud band nor do we confidently classify them as ice or liquid. Future investigations and balloon launches of instruments with higher resolutions should be undertaken to better understand this observation, but that is outside the scope of this paper. We have included a statement to this effect in the paragraph beginning at:

Line 295: "As with each of the…"

The particles in this cloud band are close in diameter to the instrument resolution limit and we associate this with the observed 'blurring' in the images. We do not expect blurring due to particle motion, as argued in response to one of the minor issues below. We agree that significant errors in particle sizing may occur when hand tracing particles. Given that these particles are close in diameter to the resolution limit of the instrument, we consider this to be the dominant challenge in deriving reliable particle sizes and this is a limitation for any method of diameter retrieval. This matter is now better addressed in the paragraph beginning at:

Line 97: "We use a diode laser with a…"

**Minor issues:**

Line 49: "Similarly, satellite based remote sensing offers wide geographical coverage". This sentence is not similar to the previous lines and ends abruptly. Please expand on the advantages and disadvantages of satellite measurements.

*Response:*

We have reworded this sentence and elaborated on the advantages and disadvantages of satellite measurements in the following paragraph beginning at:

Line 38: "Remote sensing, with lidars, radars,…"

Line 51: Please expand how installing instrumentation on aircrafts is complex due to aviation regulations and engineering difficulties.

*Response:*

We have expanded on these complexities in the paragraph beginning at:

Line 53: "Deploying instruments on aircraft requires…"

Line 88: Why limit the particle motion to 1 m/s? Falling drizzle drops move at 4 m/s downwards while the balloon ascent rate is 4 m/s upwards. Is a relative speed of 8 m/s accounted for by the pulse width without generating streaks?

*Response:*

Our original discussion of this matter was imprecise. We have updated the manuscript to include the equation used to describe this design constraint and provided an example calculation that motivated our choice of pulse width.

We can use this equation to show that the example of a falling drizzle droplet should not produce streaking with our design. Assuming a typical drizzle droplet diameter of 200~microns and a relative speed of 8 m/s, a desired particle streaking of no more than 10% of the particle diameter would require a pulse width of 2.5 microseconds or less. Our chosen laser pulse width of 100 nanoseconds is therefore sufficient to account for this issue.

The observation of circular/symmetrical particles (ie. not elongated from potential streaking) in this launch, along with the lack of streaking observed under laboratory testing at a range of relative velocities, suggests that the chosen pulse width was sufficient for the application.

These changes are included in the paragraph beginning at:

Line 97: "We use a diode laser with a…"

Line 89: 405 nm is in the Ultraviolet range.

*Response:*

We disagree with this statement and note that 405 nm falls within conventional definitions of visible light (eg. "from 400 nm to 700 nm" as in: Joint Technical Committee SF-019 (2011)), rather than Ultraviolet (UV).

UV light is not eye-safe and eventually affects lenses, glass windows and camera detectors due to high power (Spuler & Fugal, 2011).

***Response:***

We additionally disagree with these statements. Eye safety depends on many factors (see eg. Joint Technical Committee SF-019 (2011)) and the assessment of eye safety for a laser with a given optical frequency must be determined according to factors such as the average power, pulse rate, beam diameter and focussing, and various other considerations.

The power (energy/time) is an entirely different notion to the frequency of light (whether it is visible, UV, etc.). Therefore, a "high power", however that may be defined, is not a fundamental property of UV light.

For our system we use a 405 nm laser with pulses of approximately 100 ns duration and around 4 nJ of energy per pulse. The pulse rate is 1 Hz and the beam is collimated to a diameter of approximately 1 cm. We determine that this is a Class 1 laser according to the classification procedure defined in Joint Technical Committee SF-019 (2011) and can therefore be considered as eye safe. Furthermore, the system design is such that it is impossible for the eye to be directly positioned within the beam path. The laser can therefore be classified as Class 1, it can be considered eye safe, and degradation to lenses, glass windows, and the camera is negligible over any expected duration of operation. We have updated the manuscript to emphasise this eye safe design in the paragraph beginning at:

Line 97: "We use a diode laser with a…"

UV coating installed on any of the lenses/glass windows? I see that a 0.9 ND filter was installed in front of the camera. Any analysis which suggests that this value is enough to protect the detector?

***Response:***

We are particularly confused by this question. There are of course no UV coatings on our optical surfaces, as the basic operating principle of the instrument requires the efficient transfer of energy (UV or otherwise) from the laser to the camera through each of the optical elements in the system.

The 0.9 ND filter was used solely to reduce the amount of sunlight reaching the camera sensor to avoid intensity saturation. This was partially successful for this launch, though as indicated in Figure 3 it will be necessary to implement improved sunlight mitigation techniques in future launches eg. greater ND attenuation, increased camera to window spacing, etc. As outlined in our previous response, there is no need to protect the detector from the laser light (quite the opposite in fact due to the basic operating principle!).

Figure 5: Please increase the size of the histograms. At current level, the legends cannot be read. Reducing the white space will help make it a better figure with each subplot labeled a-d.

*Response:*

We have implemented these suggestions.

Line 195: Please provide the average number of particles detected per hologram in order to determine if the distributions were statistically significant.

*Response:*

The average number of particles per hologram depends on the underlying cloud particle number density, which varied significantly throughout the flight. The average value from all cloud bands (excluding holograms with no particles) was approximately 2 particles per hologram, which corresponds to an average of 60 +/- 8 (Poisson counting uncertainty) particles per 30 seconds. We have updated the manuscript text regarding this matter in the paragraph beginning at:

Line 217: "An example of manually analysed..."

Line 245: Formation of ice at these temps is actually rare since most Ice nucleating particles (INPs) activate below -8*C Kanji et al., 2017, McCluskey et al., 2019; Vergara-Temprado et al., 2018)

*Response:*

We have included a statement regarding this point along with the provided references in the paragraph beginning at:

Line 277: "Visual inspection of the cloud..."

Line 261: Any explanations why Band 3 and Band 2 particles look similar?

*Response:*

As noted in our previous response, since the particles in these cloud bands are close to the resolution limit of the holographic imager we do not wish to speculate as to why they appear to be similar. It is certainly an interesting observation, and we comment in the manuscript that this observation is noteworthy, but further investigation is outside the scope of this paper. We have now elaborated on this statement in the paragraph beginning at:

Line 295: "As with each of the..."

Line 352: Not sure where to look for small particles or low and mid-level clouds in Fig. 14.

*Response:*

We have included an additional figure indicating the cloud top heights retrieved from the satellite to clarify these issues in interpretation. The text has been updated to more carefully explain these observations in the paragraph starting at:

Line 379: "Pixels in the satellite…"

Figure 14: What are the grey pixels? Are they no-cloud pixels, or no-data pixels. How are they accounted for when determining effective radius and CTH?

*Response:*

We have updated this figure to resolve these points of confusion. The original figure showed the CER overlaid on an RGB image (converted to grayscale) from the satellite to provide additional structure in the field of view. We agree that the inclusion of this grayscale image is confusing and have decided to remove it.

White pixels in the updated figure indicate regions for which the satellite data products either do not identify cloud or do not report the particular metric for that region due to limitations of the retrieval algorithms. These regions are excluded when computing averaged quantities in this study. This matter is now clarified in the manuscript in the paragraph beginning at:

Line 379: "Pixels in the satellite…"

**References:**

Korolev, A., Isaac, G.A., 2006. Relative Humidity in Liquid, Mixed-Phase, and Ice Clouds. Journal of the Atmospheric Sciences 63, 2865–2880. https://doi.org/10.1175/JAS3784.1

Schuyler, T.J.; Gohari, S.M.I.; Pundsack, G.; Berchoff, D.; Guzman, M.I. Using a Balloon-Launched Unmanned Glider to Validate Real-Time WRF Modeling. *Sensors* **2019**, *19*, 1914. https://doi.org/10.3390/s19081914

Ramelli, F., Beck, A., Henneberger, J., Lohmann, U., 2020. Using a holographic imager on a tethered balloon system for microphysical observations of boundary layer clouds. Atmospheric Measurement Techniques 13, 925–939. https://doi.org/10.5194/amt-13-925-2020

Nash, J., et al. "WMO intercomparison of high quality radiosonde systems, Yangjiang, China, 12 July–3 August 2010." *World Meteorological Organization, Instruments and Observing methods, report No* 107 (2011), 249 pp.

Joint Technical Committee SF-019, 2011. Safety of Laser Products Part 1: Equipment Classification and Requirements, Council of Standards Australia and Council of Standards New Zealand

University of Wyoming: 94672 YPAD Adelaide Airport Observations at 00Z 08 Aug 2020, https://weather.uwyo.edu/cgi-bin/sounding?region=pac&TYPE=TEXT%3ALIST&YEAR=2020&MONTH=08&FROM=0800&TO=0812&STNM=94672&ICE=1&REPLOT=1, accessed: 2024-03-12, 2020.